The montane trees of the Cameroon Highlands, West-Central Africa, with Deinbollia onanae sp. nov. (Sapindaceae), a new primate-dispersed, Endangered species

Cheek Martin m.cheek@kew.org 1
Onana Jean Michel 2 3
Chapman Hazel M. 4
1 Department of Science, Royal Botanic Gardens, Kew , Richmond , Surrey , United Kingdom
2 Faculty of Science, Department of Plant Biology, University of Yaoundé I , Yaoundé , Centrale , Cameroon
3 IRAD, IRAD-National Herbarium of Cameroon , Yaoundé , Centrale , Cameroon
4 School of Biological Sciences, University of Canterbury , Christchurch , Canterbury , New Zealand
Casazza Gabriele
Electronic publication date: 2021 Mar 15
Publication date: 2021
Volume: 9
Electronic Location ID: e11036
Received 2020 Nov 26; Accepted 2021 Feb 9
Copyright: ©2021 Cheek et al.
Copyright year: 2021
Copyright holder: Cheek et al.
License: This is an open access article distributed under the terms of the Creative Commons Attribution License, which permits unrestricted use, distribution, reproduction and adaptation in any medium and for any purpose provided that it is properly attributed. For attribution, the original author(s), title, publication source (PeerJ) and either DOI or URL of the article must be cited.
License URL: https://creativecommons.org/licenses/by/4.0/

Keywords: Monoecious, High altitude, Forest clearance, Litchi group, Seed dispersal, Chimpanzee dispersal, Putty-nose monkey dispersal, Medicinal plant, Severe habitat fragmentation, Nigeria

Funding: Players of Peoples Postcode Lottery Bentham-Moxon Trust of RBG, Kew This paper was completed as part of the Cameroon Tropical Important Plant Areas Project, supported by Players of Peoples Postcode Lottery. Jean Michel Onana’s contribution to this paper was made possible by visits from Cameroon to RBG, Kew, U.K. sponsored by the Bentham-Moxon Trust of RBG, Kew. The funders had no role in study design, data collection and analysis, decision to publish, or preparation of the manuscript.

==============================
We test the hypothesis that the tree species previously known as Deinbollia sp. 2. is a new species for science. We formally characterise and name this species as Deinbollia onanae (Sapindaceae-Litchi clade) and we discuss it in the context of the assemblage of montane tree species in the Cameroon Highlands of West-Central Africa. The new species is a shade-bearing, non-pioneer understorey forest tree species reaching 15 m high and a trunk diameter that can attain over 40 cm at 1.3 m above the ground. Seed dispersal has been recorded by chimpanzees (Pan troglodytes ellioti) and by putty-nose monkeys (Cercopithecus nictitans) and the species is used by chimpanzees for nesting. Cameroon has the highest species-diversity and species endemism known in this African-Western Indian Ocean genus of 42, mainly lowland species. Deinbollia onanae is an infrequent tree species known from six locations in surviving islands of montane (sometimes also upper submontane) forest along the line of the Cameroon Highlands, including one at Ngel Nyaki in Mambilla, Nigeria. Deinbollia onanae is here assessed as Endangered according to the IUCN 2012 standard, threatened by severe fragmentation of its mountain forest habitat due to extensive and ongoing clearance for agriculture. The majority of the 28 tree species of montane forest (above 2000 m alt.) in the Cameroon Highlands are also widespread in East African mountains (i.e. are Afromontane wide). Deinbollia onanae is one of only seven species known to be endemic (globally restricted to) these highlands. It is postulated that this new species is morphologically closest to Deinbollia oreophila, a frequent species at a lower (submontane) altitudinal band of the same range. Detailed ecological data on Deinbollia onanae from the Nigerian location, Ngel Nyaki, where it has been known under the name Deinbollia “pinnata”, is reviewed.

Introduction

As part of the project to designate Important Plant Areas (IPAs) in Cameroon (also known as Tropical Important Plant Areas or TIPAs, https://www.kew.org/science/our-science/projects/tropical-important-plant-areas-cameroon), we are striving to name, assess the conservation status and include in IPAs (Darbyshire et al., 2017) rare and threatened plant species in the threatened natural habitat of the Cross-Sanaga interval (Cheek et al., 2001).

Several of these species were previously designated as new to science but not formally published in a series of checklists (see below) ranging over much of the Cross-Sanaga interval. The Cross-Sanaga has the highest vascular plant species diversity per degree square in tropical Africa (Barthlott, Lauer & Placke, 1996) but natural habitat is being steadily cleared, predominantly for agriculture.

In this paper we test the hypothesis that the high-altitude tree species formerly designated as “Deinbollia sp. 2” (Harvey et al., 2004; Cheek et al., 2004; Cheek, Harvey & Onana, 2010), “Deinbollia sp.” (Chapman & Chapman, 2001) or “Deinbollia pinnata” (Abiem et al., 2020), is a new species to science, and we describe, characterise, and name it as Deinbollia onanae Cheek. The species is discussed in the context of the assemblage of the other montane forest tree species (occurring above 2000 m alt.) in the Cameroon Highlands (see Discussion below).

The genus Deinbollia Schum. & Thonn. is traditionally placed in the tribe Sapindeae DC. and is characterised by its 1-pinnate, imparipinnate leaves, flowers with petals well developed and about the same in number as the imbricate sepals, the petals with a well-developed ligule (or appendage) on the adaxial surface and with stamens 9–30 in number, the intrastaminal disc central, the edge with more than 5 shallow ridges. The fruits develop 1–3 indehiscent, apocarpous fleshy mericarps (Fouilloy & Hallé, 1973).

Molecular phylogenetic sampling of the Sapindaceae is incomplete with many African genera not represented, as can be seen in Buerki et al. (2009). In that study Deinbollia is represented by six samples of four species, all from Madagascar (on which limited basis it appears monophyletic) and is resolved in the informally named ‘Litchi Group’ of genera, where it is in a sister relationship to a subclade comprising the genera Lepisanthes Blume (Africa to Asia) Atalaya Blume (American) and Pseudima Radlk. (American) (Buerki et al., 2009). The delimitation of Sapindaceae in this paper follows the evidence of Buerki et al. (2010), that is, excluding Aceraceae, Hippocastanaceae and Xanthoceraceae which have sometimes been included within it.

Deinbollia has 42 accepted species, one shared between Africa, Reunion and Madagascar, five endemic to Madagascar, and 35 species restricted to subsaharan continental Africa. The species predominantly occur in lowland evergreen forest and are absent from countries that lack this habitat such as Rwanda, Burundi, Swaziland, Lesotho (high altitude), Namibia, Botswana, Eritrea, Mali and Burkina Faso (low rainfall and lacking significant evergreen forest). The highest species diversity is found in Cameroon, with 16 species (Plants of the World Online accessed May 2020). Cameroon has the highest levels of country-level endemism known in the genus. Ten of the Cameroon species are globally threatened with extinction (Cheek in Onana & Cheek, 2011: 314–316; Cheek, 2004; Cheek, 2017a; Cheek, 2017b). In contrast only 10 species are recorded for the whole of West Tropical Africa (Keay, 1958). Since the Flore du Cameroun account was published (Fouilloy & Hallé, 1973), several further species apart from those listed below, were published for Cameroon by Thomas (1986). The genus was last revised by Radlkofer (1932).

In the 21st century only two new species to science have been published in the genus, Deinbollia mezilii Thomas & Harris (2000) and D. oreophila Cheek (Cheek & Etuge, 2009a), both from Cameroon. But specimens often remain unidentified in herbaria. For example, 16 specimens unidentified to species are listed in the Gabon Checklist (Sosef et al., 2005). The genus has no major uses, but the fruits of several species are reported as being edible by humans, and the seeds are probably primate-dispersed or dispersed by large frugivorous birds, and the flowers probably bee-pollinated (Cheek & Etuge, 2009a; Cheek & Etuge, 2009b). Several species are recorded to be useful locally in West Africa especially medicinally, e.g., the bark of D. grandifolia Hook.f. is used for treating jaundice and the wood for planks (Burkill, 2000: 17–19).

Methods & Materials

The electronic version of this article in Portable Document Format (PDF) will represent a published work according to the International Code of Nomenclature for algae, fungi, and plants (ICN), and hence the new names contained in the electronic version are effectively published under that Code from the electronic edition alone. In addition, new names contained in this work which have been issued with identifiers by IPNI will eventually be made available to the Global Names Index. The IPNI LSIDs can be resolved and the associated information viewed through any standard web browser by appending the LSID contained in this publication to the prefix “http://ipni.org/”. The online version of this work is archived and available from the following digital repositories: PeerJ, PubMed Central, and CLOCKSS.

Fieldwork in Cameroon resulting in the specimens cited in this paper was conducted under the terms of the series of Memoranda of Collaboration between IRAD (Institute for Agronomic Research and Development)-National Herbarium of Cameroon and Royal Botanic Gardens, Kew beginning in 1992, the most recent of which is valid until 5th Sept. 2021. The most recent research permit issued for fieldwork under these agreements was 000146/MINRESI/B00/C00/C10/C12 (issued 28 Nov 2019), and the export permit number was 098/IRAD/DG/CRRA-NK/SSRB/12/2019 (issued 19 Dec 2019). At the Royal Botanic Gardens, Kew, fieldwork was approved by the Institutional Review Board of Kew entitled the Overseas Fieldwork Committee (OFC) for which the most recent registration number was OFC 807-3 (2019). The most complete set of duplicates for all specimens made was deposited at YA, the remainder exported to K for identification and distribution following standard practice. Field work methodology followed was (Cheek & Cable, 1997). Herbarium citations follow Index Herbariorum (Thiers, 2020). Specimens indicated “!” were seen by one or more authors, and were studied at K, P, WAG, and YA. The National Herbarium of Cameroon, YA, was also searched for additional material of the new taxon as was Tropicos (http://legacy.tropicos.org/SpecimenSearch.aspx). During the time that this paper was researched in 2019–2020, it was not possible to obtain physical access to material at WAG (due to the transfer of WAG to Naturalis, Leiden, subsequent construction work, and covid-19 travel and access restrictions). However images for WAG specimens were studied at https://bioportal.naturalis.nl/?language=en and those from P at https://science.mnhn.fr/institution/mnhn/collection/p/item/search/form?lang=en_US. Specimens of Deinbollia at FHO could not be accessed due to covid-19 and are not available digitally. Specimens at FHI are also not available digitally. We also searched JStor Global Plants (2020) for additional type material of the genus not already represented at K.

Binomial authorities follow the International Plant Names Index (IPNI, 2020). The conservation assessment was made using the categories and criteria of IUCN (2012). GeoCAT was used to calculate red list metrics (Bachman et al., 2011). Herbarium material was examined with a Leica Wild M8 dissecting binocular microscope fitted with an eyepiece graticule measuring in units of 0.025 mm at maximum magnification. The drawing was made with the same equipment using Leica 308700 camera lucida attachment. Flowers from herbarium specimens of the new species described below were soaked in warm water to rehydrate the flowers, allowing dissection, characterisation, and measurement. The terms and format of the description follow the conventions of Cheek & Etuge (2009a); Cheek & Etuge (2009b). Georeferences for specimens lacking latitude and longitude were obtained using Google Earth (https://www.google.com/intl/en_uk/earth/versions/). The map was made using SimpleMappr (https://www.simplemappr.net).

Results

Taxonomic treatment

Deinbollia sp. 2 (Fig. 1), because it has leaves of flowering branches less than 1 m long, only sparsely hairy on the lower surface, leaflets more than 15 cm long and sepals adaxially glabrous, flower buds very sparsely hairy and less than 5 mm diam., borne on a branched inflorescence 10–30 cm long, keys out in the Flore du Cameroun treatment of Deinbollia (Fouilloy & Hallé, 1973) to a couplet leading to D. grandifolia Hook.f. and D. maxima Gilg. However, it differs from these two species in having (2–)8–11-jugate leaves (not 4–7-jugate), and in other characters shown in Table 1. In its Nigerian location our species has been referred to as D. pinnata (Abiem et al., 2020). Deinbollia pinnata Schum. & Thonn. is a common lowland West African species that occurs from Guinea to Nigeria, it differs in being densely hairy, so that the lower surface of the leaflets are softly hairy to the touch due to dense, patent, translucent hairs, and it is usually a small shrub of disturbed habitats, with an unbranched, raceme-like inflorescence that is pendulous in fruit, with hairy fruits 12–13 mm wide (see https://commons.wikimedia.org/wiki/File:Deinbollia_pinnata_MS6765.jpg; Keay, 1958: 714–715). In contrast, Deinbollia sp. 2 has only a very few, sparse, red, subappressed hairs along the midrib and secondary nerves, is a tree of intact high elevation forest, the inflorescence is erect, with numerous long branches bearing glabrous fruits 20 mm or more wide (see description below). Additional characters separating Deinbollia sp. 2 from Deinbollia pinnata are included in Table 1.

Figure 1 Deinbollia onanae.

(A) Habit, flowering branch; (B) detail from a large leaf showing apex and distal leaves (adaxial surfaces) and second leaf from the base (abaxial surface); (C) male flower lateral view; (D)male flower, petals and sepals removed to show the extra staminal disc and androecium; (E) base of D (male flower) showing the vestigial gynoecium and disc cut to show notches holding filament bases; (F) petal, adaxial surface, male flower; (G) female flower, lateral view; (H) female flower (with 3 sepals, 2 petals and anterior stamens removed) to show gynoecium and disc; (I) stamen from female flower; (J) petal, adaxial surface, of female flower with stamen. A, C-J from de Wilde et al. 4553 (K); B from Cheek 13436 (K). Drawn by Andrew Brown.

The affinities of Deinbollia sp. 2 may be with the recently described D. oreophila since this species also occurs at altitude in the Cameroon Highlands and both species share numerous raised lenticels and also leaflets with high length: breadth ratios and with high numbers of secondary nerves. Both species share an unusual structure which is also seen in Deinbollia pinnata: the adaxial surface of the leaf rhachis is not rounded as in the other West African species, but flattened, the margins slightly raised forming acute angles with the sides, with a distinct, raised midline (Cheek & Etuge, 2009a: Fig. 1C). In fact, at two locations, Mt Kupe and Bali Ngemba, the two species D. oreophila and Deinbollia sp. 2 are sympatric and their altitudinal ranges can overlap (Cheek et al., 2004; Harvey et al., 2004). However, without DNA studies, convergent evolution cannot be ruled out. As the only two species of the genus to grow at altitude in the Cameroon Highlands, there is a possibility that they might be confused with each other. The two species can be separated using the morphological characters presented in Table 2.

Deinbollia onanaeCheek sp. nov. –Figs. 1–4	

Similar to but differing from Deinbollia oreophila Cheek in the length of leaves of flowering stems (14–)60–70 cm (versus 25–63 cm), number of leaflets per leaf (4–)16–23 (versus (4–)6–8(–10)), width of leaflets (2.1–)2.5–4 cm (versus (3–)5.5–9(–10.2) cm, number of secondary nerves on each side of midrib (15–)17–18, (versus (7–)9–14(–17); stems with lenticels brown, concolorous and inconspicuous, (versus discolorous, bright white and conspicuous), ovary bilocular (versus trilocular).Typus: Cameroon, Mt Oku and the Ijim Ridge, Aboh to Tum, 2400 m alt., fl. 22 Nov. 1996, Etuge 3600 (holotype K000337729! Fig. 2, isotypes MO!, WAG0336084!, WAG0336083!, YA0057050!);

Deinbollia cf. pinnata Schum. & Thonn., sensu Cheek, in Cheek, Onana & Pollard (2000:162).	
Deinbollia sp. 2 sensu Cheek in Harvey et al. (2004: 125); Cheek & Etuge in Cheek et al. (2004: 399); Cheek in Cheek, Harvey & Onana (2010: 143, fig 23).	
Deinbollia sp. Chapman & Chapman (Chapman & Chapman, 2001: c41)	

Table 1 Characters separating Deinbollia onanae from D. grandifolia, D. maxima and D. pinnata.

Characters taken from Fouilloy & Hallé (1973) and Keay (1958).

	Deinbollia grandifolia	Deinbollia maxima	Deinbollia onanae	Deinbollia pinnata	
Leaves	(5–)7-10-jugate	4–6-jugate	(2–)8–11-jugate	(2–)5–9-(-12)-jugate	
Leaf rhachis adaxial surface	Rounded	Rounded	Flattened, with margins angled-winged, midline with raised ridge	Flattened, with margins angled-winged, midline with raised ridge	
Indumentum of abaxial surface of leaflet	Glabrous, or with a few scattered inconspicuous hairs	Glabrous, or with a few scattered inconspicuous hairs	Glabrous, or with a few scattered inconspicuous hairs	Softly hairy with numerous translucent, patent hairs	
Leaflet width	5–8 cm	6–8(–10) cm	(2.1–)2.5–4 cm	2.3–7.5(-10) cm	
N∘s pair of secondary nerves (distal leaflets)	15–14(–16)	8 –10	(12–)17–18	6–12	
Fruit breadth, indumentum	1.5 cm, glabresecent	Dimensions unknown, glabresecent	2 cm, glabrous	1.3–1.5 cm, tomentose	

Table 2 The more significant differences between Deinbollia onanae and Deinbollia oreophila.

Data on Deinbollia oreophila from Cheek & Etuge (2009a).

	Deinbollia oreophila	Deinbollia onanae	
Height at maturing	0.8–3(–5) m	(4–)5–10(–15) m	
Stem indumentum	Glabrous	Simple hairy, sparse to dense, glabrescent.	
Lenticels	Highly conspicuous, bright white, contrasting with epidermis	Inconspicuous, grey-brown, concolorous with epidermis	
Length of leaves (flowering stems)	25–63 cm	(14–)60–70 cm	
Number of leaflets per leaf (flowering stems)	(4–)6–8(–10)	(4–)16–23	
Width of leaflets (flowering stems).	(3–)5.5–9(–10.2) cm	(2.1–)2.5–4 cm	
N ∘ secondary nerves each side of midrib	(7–)9–14(–17)	(15–)17–18	
Indumentum of lower surface of leaf blade	Glabrous	Inconspicuously sparsely simple hairy on secondary nerves and midrib	
Sepals	Orbicular, margins glabrous	Ovate, margins hairy	
Petals	Oblong or obovate, base cuneate; adaxial appendage surface glabrous	Rhombic or spatulate, basal claw (stalk); adaxial appendage surface hairy	
Staminal filaments of male flowers	Proximal half glabrous.	Entire length densely hairy.	
Ovary of female flowers	3-lobed, surface with very sparse, stout hairs	2-lobed, densely hairy with fine hairs	
Altitudinal range	(880–)1000–2050 m	(1400–)2050–2200 m	

Figure 2 Deinbollia onanae.

Photo of the holotype: Etuge 3600 (holotypus K000593309). Photo by Xander van der Burgt.

Figure 3 Deinbollia onanae. Global distribution map.

By Xander van der Burgt.

Figure 4 Deinbollia onanae.

Habit of tree in flower at Ngel Nyaki, Nigeria. Photo by Max Walters. Source: Nickrent et al. (2006).

Monoecious tree or treelet (4–)5–15 m tall, trunk 14.5–40 cm diameter at 1.3 m from the ground, lacking exudate or scent when wounded, sparingly branched, nearly glabrous, apart from the inflorescence. Stems of flowering branches terete (0.8–)1–1.5 cm diameter, solid (not hollow), second internode below apical inflorescence 2–2.5 cm long, outer epidermis pale grey-brown, contrasting with the darker brown bases of the adjoining petiolar pulvini, lenticels dense, raised, elliptic, 0.6–1.1 mm long, concolorous, inconspicuous, glabrescent, hairs sparse to dense, dark brown, cylindric 0.1–0.5 mm long.

Leaves alternate, pinnately compound, (14–)60–70 cm long; leaflets (4–)16–23 per leaf on flowering stems, leaflets 10–14 per leaf on leaves of juvenile trees. Petiole (4–)9.5–20.8 cm long, terete, c. four mm diameter at midpoint, drying pale yellow; basal pulvini dark brown; rhachis (4.5–)32–44 cm long, (2–)8–11-jugate on flowering stems, 5–7-jugate on non-flowering stems of juvenile trees, the upper surface of the distal half flattened with two thin lateral wings and with a central dark hairy rounded central ridge, the rest of the rhachis glabrescent with sparse inconspicuous hairs (de Wilde 4555), or with sparse dark brown appressed hairs (Cable 3386). Leaflets mostly oblong (6.6–)14–19.5 × (2.1–)2.5–4 cm, (but leaflets of sterile branches to 6.5 cm wide), acumen c. 1 cm long, base broadly acute, slightly asymmetric, (basalmost leaflets lanceolate and about half the length of the other leaflets) lateral nerves and midrib yellow, raised above and below, convex, (15–)17–18 on each side of the midrib, nearly brochidodromous, the lateral nerve apices forming a weak irregular submarginal nerve, stronger branches uniting with the secondary nerve above, intersecondary nerves strong, parallel to the secondaries, tertiary and quaternary nerves reticulate raised yellow and conspicuous, on both surfaces, contrasting with the pale grey-green areolae (except in Cable 3386(K) where they are concolorous and so inconspicuous above, possibly an artefact of poor drying); upper surface glabrous, lower surface with inconspicuous, minute, cylindrical, subappressed glossy dark-brown hairs c. 0.25 mm long, distributed very sparsely along the midrib and secondary nerves, absent from mature leaves of non-flowering specimens (e.g., Cheek 8709) but then the same hair type present on axillary buds and young leaves; petiolules yellow, 2–5 mm long, glabrous.

Inflorescence a 80–120-flowered, loose, terminal panicle 25 × 10 cm; auxiliary inflorescences sometimes present in the axils of the distal 1–4 leaves (Cheek 13625); peduncle of terminal inflorescences 0–2 cm long; rhachis internodes (1–)2–3 cm long, shortest in the distal portion; first order bracts caducous; indumentum brown hairy; primary branches 10–20 per inflorescence, 2–8 cm long, each bearing (1–)2–5 partial-inflorescences; partial-peduncles 0–5 mm long, apex with a cluster of 3–5 bracteoles; bracteoles subulate to narrowly lanceolate, 2–3 mm long, apex narrowly acute, partial-inflorescences (1–)3-flowered in glomerules, pedicels erect, terete, 3–4 × 1.5 mm (female), 4–5 × 1 mm (male), sparsely puberulent, hairs 0.1–0.5 mm long.

Flowers white, scent not recorded, flower buds c. four mm diam., open flowers c. 6 × 7 mm. Calyx with sepals 5(–6), orbicular to broadly ovate, concave, green colour, 4–5 × 3.5–4.5 mm apex obtuse. Corolla apex slightly exserted from calyx, petals rhombic or spatulate. Male flowers (Fig. 1C). Petals 5(–6), white, rhombic c. 5 × 3 mm, apex obtuse-acute, base cuneate, margins densely ciliate, hairs 0.3 mm long, outer surface glabrous, inner surface glabrous in distal half, proximal half compressed funneliform with ventral appendage adnate at margins, retuse (notched) for 0.5 mm at midline, adaxial surface moderately densely hairy, hairs c. 0.3 mm long. Extra-staminal disc torus-like, glabrous, irregular, outer wall convex, lacking constrictions or teeth with c. 15 poorly defined lobes, 2.5–3 mm wide, c. 0.8 mm high. Stamens c. 15, erect, slightly exserted by 1–2 mm at anthesis, c. 5–6.5 mm long; filament 4–5 mm long, straight, densely puberulent the entire length (Fig. 1D); anthers yellow, ovate-ellipsoid, 1–1.3 mm long. Ovary (vestigial, Fig. 1E) bilobed, c. 1 × 1.5 mm densely appressed hairy, hairs c. 0.5 mm; style 0.7 mm long, glabrous.

Female flowers (Fig. 1G), with sepals and petals as the male flowers, but petals c. 6 × 2.6–2.9 mm, usually detaching with a stamen attached, probably due to interlocking hairs (see Fig. 1J), proximal two-thirds claw-like, c. 0.7 mm wide, margin sparsely and irregularly ciliate; ventral appendage with apex deeply bilobed, lobes c. 1 mm × 1 mm; disc as in male flower. Stamens c. 10 (see Fig. 1I), included at anthesis, filament c. 2.5 mm long, proximal half to quarter glabrous, distal part densely hairy; anther as male flowers but indehiscent; ovary bilobed (see Fig. 1H), 3.2  × 5 mm, indumentum as male flower, style c. 5 mm long, apical 1 mm, curved, surface papillate-minutely puberulent, apex subcapitate. Infructescence, of same dimensions as inflorescence, erect. Fruit colour recorded as nearly black when ripe, tasting sweet-sour (Elisha Barde, see uses below), and not coloured yellow when ripe (as in other species of the genus), mericarps 1 or 2, transversely ellipsoid, c.1.8 × 2.1 × 1.2 cm (hydrated), the surface leathery, shallowly and finely muricated, glabrous, mesocarp spongy and juicy, 1-seeded. Seed ellipsoid, c. 1.8 × 1.1 × 0.8 cm, testa thin, parchment like, endosperm absent, cotyledons fleshy.

Phenology: flowering in November-December; fruiting in February and April, immature fruit recorded in December and June.

Local name and uses: none are reported in Cameroon but in Ngel Nyaki, Nigeria, Elisha Emmanuel Barde of the Nigeria Montane Forest Project (pers. comm. to M. Cheek Dec. 2020), states that Nyeberehi (Fulfude) is the general name for all Deinbollia species while Jellahi (Fulfude) is a specific name for Deinbollia onanae in Ngel Nyaki where Fulfude speakers (Fulanis) use the bark of this species as medicine for themselves, to treat stomach aches as well as an anti-helminthic. It is not used for treating cattle. The fruits are reported to taste sour-sweet by Mr Barde. The species is also known as Pabba (Ndolla language).

Etymology: The specific epithet of Deinbollia onanae means ‘of Onana’ commemorating Dr Jean-Michel Onana, currently Senior Lecturer in Botany at the University of Yaoundé I, Cameroon, champion of plant conservation in Cameroon, specialist in Sapindales (Burseraceae, author of Flore du Cameroun Burseraceae (Onana, 2017), co-chair of the IUCN Central African Red List Authority for Plants, former Head of the National Herbarium of Cameroon (2005–2016), co-author of the Red Data Book of the Plants of Cameroon (Onana & Cheek, 2011) and the Taxonomic Checklist of the Vascular Plants of Cameroon (Onana, 2011). He led field teams of YA staff working with those of K that resulted in the collection of several of the specimens of this species and personally collected this species in the field (Onana 1600, K, YA).

Distribution & ecology: known only from the Cameroon Highlands of Cameroon (one location in the adjoining Mambilla Plateau, Nigeria) Fig. 3. Upper submontane & montane evergreen forest, sometimes in gallery forest; (1200–)2,050–2,200 m alt.

Additional specimens: CAMEROON. South West Region, Mt Kupe, near main summit, immature fr., 26 June 1996, Cable 3386 (K000197863!, YA!); North West Region.

Bali Ngemba Forest Reserve, fr. April 2002, Onana 1600 (K!); Mt Oku and the Ijim Ridge: above Laikom, st. 21 Nov..1996, Cheek 8709 (K000337728! YA!); Dom, Kinjinjang Rock, st. 25 Sept. 2006, Cheek 13436 (K000580433!; YA!); ibid. Forest Patch 1, fl. buds, 27 Sept. 2006, Cheek 13625 (K000580434!, MO!,US!, YA!); ibid., Javelong Forest, st. 29 April 2005, Pollard 1400 (K000580432!; YA!); Adamaoua Region, c. 120 km E of Ngaoundéré, 15 km NE of Belel, falls in Koudini River, alt. ± 1200 m, fl. 4 Dec. 1964, W.J.J.O. & J.J.F.E. de Wilde, B.E.E. de Wilde-Duyfjes 4555 (K000593309!; K000593310!, WAG1269760!, YA). NIGERIA. Taraba State, Mambilla Plateau, Ngel Nyaki Forest Reserve, near camp, fr. 2 Dec. 2003, H.M. Chapman 481 (FHI, K!); ibid. female fl. 4 Dec. 2002, H.M. Chapman 484 (FHI, K!).

Notes: Deinbollia onanae first came to our attention in 2000 when completing the “Plants of Kilum-Ijim” (Cheek, Onana & Pollard, 2000). Two specimens of Deinbollia matched no other and were named Deinbollia cf. pinnata (Cheek, Onana & Pollard, 2000). In subsequent surveys this taxon was more explicitly referred to as a new species: Deinbollia sp. 2 (Harvey et al., 2004; Cheek et al., 2004; Cheek, Corcoran & Horwath, 2009). However, the earliest known collection was made in 1964 (W.J.J.O. & J.J.F.E. de Wilde, de Wilde-Duyfjes 4555(K)).

This species is remarkable for the very large number of pairs of unusually long and slender leaflets (Fig. 4), and for the comparatively large size of the individuals which often attain 10–15 m in height (Fig. 4), among the largest trees known in the genus. However, at Ngel Nyaki trees can begin flowering at only 2.5 m in height (E. Barde pers. comm. to Cheek Jan. 2020)

Conservation: Deinbollia onanae is rare at each of its six known locations so far as is known, although at Ngel Nyaki this is difficult to establish due to potential confusion with Deinbollia oreophila. Despite many thousands of herbarium specimens being collected at Kilum-Ijim, at Mt Kupe and the Bakossi Mts, at Ngel Nyaki and at Bali Ngemba (Cheek, Onana & Pollard, 2000; Cheek et al., 2004; Harvey et al., 2004) only two specimens of this species at two sites, were made at each of the first three locations and only one at the third location. Surveys at other sites with suitable habitat in the Cameroon Highlands and elsewhere, e.g at Mt Cameroon and at the Lebialem Highlands, failed to find this species (Cheek et al., 1996; Cable & Cheek, 1998; Harvey, Tchiengue & Cheek, 2010; Cheek, Harvey & Onana, 2011). However, at Dom, where a targetted search for this species was made by the first author, three specimens were made, each representing single, isolated trees Cheek, Harvey & Onana (2010). No more individuals than these were found. At Adamaoua Region, Cameroon it has only been collected once, and only a single tree was then noted (W.J.J.O. & J.J.F.E. de Wilde, B.E.E. de Wilde-Duyfjes 4555(K)). None of these locations is formally protected for nature conservation. Tree cutting for timber and habitat clearance for agriculture has long been known to be a threat at all but the last of these locations (references cited above). The range of the species is large: extent of occurrence was calculated as 50,525 km2 using GeoCAT. However, severe habitat fragmentation has resulted over many hundreds of years, forest patches being now distant from each other by tens of kilometres, isolated in oceans of cultivation and secondary fire-maintained grassland making the possibility of primate-mediated dispersal from one forest area to another now extremely unlikely. Ecological evidence from Ngel Nyaki is that while Deinbollia regenerates in that forest patch, its primate dispersers do not, or seldom cross to other forest patches (Dutton & Chapman, 2015, see discussion below). We assess the area of occupancy of Deinbollia onanae as 34 km2 using the IUCN preferred 4 km2 cell size. Therefore, we assess this species as Endangered, EN B2ab(iii) using the IUCN (2012) standard. We suggest that this species be included in forest restoration plantings within its natural range to partly reverse its move to extinction. However, the large (c. one cm diam.), thin-walled seeds are probably recalcitrant, so not suitable for conventional seed-banking, and should not be allowed to be dried before sowing since this can be expected to kill them. Experience at Ngel Nyaki (Matthesius, Chapman & Kelly, 2011) shows that it is possible to raise hundreds of seedlings in nurseries and to establish them in natural forest.

Discussion

The discovery of a threatened, new species to science from surviving natural habitat in the Cameroon Highlands is not unusual. At most of the six locations from which we here describe Deinbollia onanae, additional new or resurrected species to science, all highly localised, range-restricted and threatened with extinction, have been documented in recent years. At Ngel Nyaki in Nigeria a point endemic Memecylon species (H.M. Chapman 744) as yet undescribed is present (R.D. Stone to Hazel Chapman,pers. comm., 2007), while at Mt Kupe for example, Coffea montekupensis Stoffelen (Stoffelen et al., 1997) and more recently the new species and genus to science Kupeantha kupensis Cheek & Sonké (Cheek et al., 2018a). At Bali Ngemba, Leptonychia kamerunensis Engler & K. Krause (Cheek et al., 2013), Psychotria babatwoensis Cheek (Cheek, Corcoran & Horwath, 2009) and Allophylus ujori Cheek (Cheek & Etuge, 2009b), at Mt Oku and the Ijim Ridge Kniphofia reflexa Marais (Maisels, Cheek & Wild, 2000), Scleria cheekii Bauters (Bauters, Goetghebeur & Larridon, 2018, while at Dom, the endemic epiphytic sedge Coleochloa domensis Musaya & D.A Simpson (Muasya et al., 2010). No additional such new species are known from the Adamaoua location, probably because it is less completely sampled than the preceeding four.

However, Deinbollia onanae is exceptional among these aforementioned species in that it is a new species of tree predominantly of montane forest. The many other newly discovered for science, resurrected or rediscovered plant species of the Cameroon Highlands have been overwhelmingly either been herbs or shrubs or are derived from submontane habitats (800–2,000 m altitude).

Detailed information on the ecology of Deinbollia onanae (under the name D. pinnata) is available from several studies led by Hazel Chapman at Ngel Nyaki, the largest or one of the largest, surviving forests in the Mambilla Plateau, a branch of the Cameroon Highlands that extends into Nigeria (see map, Fig. 3). At this submontane forest patch, area c.5.7 km2, 1,588–1,690 m altitude, Deinbollia (Fig. 4) is recorded as one of the 20 most abundant woody plant species, with 158.68 stems above one cm diam. per ha (Abiem et al., 2020). In contrast, the 1970s 1 ha enumeration plot at Ngel Nyaki (Chapman & Chapman, 2001: 25–26) yielded five stems of “Deinbollia sp.” in the C strata (understorey trees 7–13 m high) with diameter at 1.3 m above the ground exceeding 14.5 cm, of which two exceeded 28 cm and one 40 cm. This is more consistent with frequencies observed in Cameroon for Deinbollia onanae. The explanation between the disparity in stem numbers per ha between these two studies is probably that there is high mortality of juveniles of Deinbollia onanae at Ngel Nyaki, few surviving to make 14.5 cm diameter or more trees recorded in the second study. We speculate that an alternative explanation may be that the numerous small diameter individuals recorded by Abiem et al. (2020) may not be the usually infrequent D. onanae, but the similar but much smaller (0.8–3(–5) m tall) D. oreophila which at this altitude, over the border in Cameroon, is vastly more frequent in submontane forest (Cheek & Etuge, 2009a). Many of the observations of animals e.g., putty-nosed monkeys (Cercopithecus nictitans) feeding on Deinbollia at Ngel Nyaki (Gawaisa, 2010) were of primates in the crowns of trees so more likely to be of the larger, less frequent D. onanae which is evidenced at this location by two herbarium specimens (see “additional specimens” above) while D. oreophila has not yet been so recorded (and so may not in fact be present). Studies on the dietary preferences of the rare Nigerian to Cameroon chimpanzee (Pan troglodytes ellioti) by Dutton & Chapman (2015) at Ngel Nyaki found that among the 52 plant species consumed mainly as fruit, Deinbollia was the 4th (wet season) or 5th (dry season) species preferred of the 17 tree species over 10 cm diameter at breast height that were identified as seeds from 495 fecal samples. This record is certainly D. onanae since D. oreophila does not form trunks of such large diameters (Cheek & Etuge, 2009a). However, Deinbollia was found in only one of these fecal samples, in which 47 of its seeds were recorded, collected in February 2011 (Dutton & Chapman, 2015). Only 16 weaned individuals of chimpanzee are known at Ngel Nyaki. More numerous and so probably more effective at seed dispersal are putty-nosed monkeys (Cercopithecus nictitans). (Gawaisa, 2010) reported that at Ngel Nyaki Deinbollia fruit ranked third in preferred species of fruits of C. nictitans during February and March, and fifth in January. Hutchinson (2015) has shown that males in particular show a preference for Deinbollia seeds in the rainy season (Hutchinson, 2015). Seeds are both swallowed, passing through the gut (average 2 per faecal sample) and sucked and spat by the putty-nosed monkeys (averaging 5 seeds per spitting event) (Chapman, Goldson & Beck, 2010). An experiment comparing germination time and success among Deinbollia seeds which had been defecated, spat and hand-cleaned found that gut passage had a significant beneficial effect on germination rates. A higher proportion of defecated seeds (60–70%) germinated than spat (c. 40%) or hand-cleaned (c.35 %) seeds, and the defecated seeds germinated on average a few days earlier than non-defecated seeds (Chapman, Goldson & Beck, 2010). In addition, leaves but not fruit of Deinbollia have been recorded as being consumed by tantalus monkeys (Chlorocebus tantalus tantalus), but only in very low quantities (Agmen, Chapman & Bawuro, 2010). Putty nose monkeys also eat the leaves and flowers of Deinbollia (Gawaisa, 2010; Hutchinson, 2015). Studies of dispersal of seeds of about 40 Ngel Nyaki forest species up to 30 m into grassland from the forest edge using seed traps showed that Deinbollia was one of the small number of forest species that do not disperse seeds out of the forest, but that within forest, natural regeneration from seed does occur. The species has been classified as a shade-bearer and is not a pioneer (Barnes & Chapman, 2014). Deinbollia “pinnata” was one of three species of tree used to test the Janzen-Connell hypothesis at this site. Five hundred and seventy seedlings were raised and planted at distances of up to 25 m from 19 mature conspecific “mother” trees and monitored over the three months of the study. Predation was significantly higher closer to the mother trees than distant from them (c. 30% vs. 20%), but there was no support for Janzen-Connell effects in seedling height growth. About 80% of the seedlings survived, and they grew 4.5–5.5 cm over the 3 months (Matthesius, Chapman & Kelly, 2011). Deinbollia “pinnata” is one of 28 identified tree species used by chimpanzees as nesting trees at Ngel Nyaki, but is not among the preferred top five (Dutton, Moltchanova & Chapman, 2016).

Table 3 The 28 montane forest trees of the Cameroon Highlands.

Data mainly from Cheek, Onana & Pollard (2000), updated with subsequent literature e.g. Kenfack (2011), Cheek & Ngolan (2007), Cheek, Tchiengue & Tacham (2017) and POWO (2019).

Currently accepted species name	Former name used in Cameroon Highlands, if any (e.g., Cheek, Onana & Pollard, 2000)	Endemic to Cameroon Highlands (Y/N)	Occurring also below 2,000 m alt. (Y/N)	Species forming 90% of the canopy (Y/N)	Forest edge species = E Infrequent species = R	
Astropanax abyssinica (A.Rich.)Seem.	Schefflera abyssinica (A.Rich.)Harms	N	N	Y		
Astropanax mannii (Hook.f.)Seem.	Schefflera mannii (Hook.f.)Harms	Y	N	Y		
Prunus africana (L.)Kalkman	Pygeum africanum Hook.f.	N	N	Y		
Syzygium staudtii (Engl.)Mildbr.	Syzygium guineense subsp. bamendae F.White	N	N	Y		
Myrsine melanophloeos (L.)Sweet	Rapanea melanophloeos (L.)Mez	N	N	Y		
Oldeania alpina (K.Schum.)Stapleton	Arundinaria alpina K.Schum.	N	N	Y		
Carapa oreophila Kenfack	Carapa grandiflora Sprague	Y	N	Y		
Bersama abyssinica Fresen.		N	N	Y		
Ixora foliosa Hiern		Y	N	Y		
Clausena anisata (Willd.)Benth,		N	Y	Y		
Nuxia congesta Fresen.		N	N	N	E	
Lasiosiphon glaucus Fresen.	Gnidia glauca (Fresen.) Gilg	N	N	N	E	
Hypericum revolutum Vahl subsp. revolutum		N	N	N	E	
Maesa lanceolata G.Don		N	N	N	E	
Alsophila dregei (Kunze)R.M.Tryon	Cyathea dregei Kunze	N	N	N	E	
Podocarpus latifolius (Thunb.)Mirb.		N	Y	N	R	
Croton macrostachyus Delile		N	Y	N	R	
Albizia gummifera (J.F.Gmel)C.A.Smith		N	Y	N	R	
Cassipourea malosana (Baker)Alston		N	N	N	R	
Brucea antidysenterica J.F.Mill.		N	N	N	R	
Ilex mitis (L.)Radlk.		N	N	N	R	
Neoboutonia mannii Benth. & Hook.f.	Neoboutonia glabrescens Prain	N	Y	N	R	
Olea capensis subsp. macrocarpa (C.H.Wright)I.Verd.	Olea capensis	N	N	N	R	
Eugenia gilgii Engl. & Brehmer		Y	N	N	R	
Agarista salicifolia (Lam.)G.Don	Agauria salicifolia (Lam.)Oliv.	N	N	N	R	
Dovyalis cameroonensis Cheek & Ngolan	Dovyalis sp.nov.	Y	N	N	R	
Ternstroemia cameroonensis Cheek	Ternstroemia polypetala Melch.	Y	Y	N	R	
Deinbollia onanae Cheek	Deinbollia sp. 2	Y	Y	N	R	

Montane Forest Trees of the Cameroon Highlands

The Cameroon Highlands extend through four tropical African countries. Beginning in the south on the volcano island of Bioko (Equatorial Guinea) they continue on the mainland with the Mount Cameroon active volcano, heading NNE along a major fault, forming the ridges, plateaux and isolated peaks of the Bakossi Mts and Mt Kupe, Muanenguba, Bamboutos Mts, the Lebialem and Bamenda Highlands, Mt Oku, Tchabal Mbabo, then heading eastwards and forming the lower and drier Adamaoua Highlands which extend into the Central African Republic. Two westward extending arms from the central section in Cameroon extend into Nigeria, forming the Obudu and Mambilla Plateaux. The altitudinal division between montane and submontane forest is well-marked in the Cameroon Highlands. Most species of montane tree only occur above the 2000 m contour and not below it (Table 3), while tree species from the submontane forest belt rarely exceed the 2000 m contour (Cheek et al., 1996; Cheek, Onana & Pollard, 2000; Cheek et al., 2004; Thomas & Cheek, 1992), although some species of tree, like Deinbollia onanae can occur on either side of the 2,000 m contour. Since most of the Cameroon Highlands do not ascend above 2,000 m alt., montane forest is not ubiquitous along their length. Moreover, even where altitude is sufficient to support it and where it formerly occurred, montane forest has seen massive clearance for agriculture, and has been totally lost at the Bamboutos Mountains of West Region Cameroon (Ngoufou, 1992). Indeed, the Bamenda Highlands of Cameroon, long since cleared of their montane forest, are now known in Cameroon as “The grasslands” because they are blanketed in secondary grassland, perpetuated by fire. It has been estimated that as much as 96.5% of the original montane forest of the Bamenda Highlands has been lost (Cheek et al., 2020, 49–50). The tallest mountain in the range, Mt Cameroon (4,040 m), despite its height and lack of human activity above 2,000 m alt., has surprisingly little forest above this contour due to the free-draining nature of its predominantly volcanic cinder substrate (Thomas & Cheek, 1992; Cheek et al., 1996; Cable & Cheek, 1998). The single largest block of montane forest that survives by far in the Cameroon Highlands is that at Mt Oku and the Ijim Ridge (Kilum-Ijim) where about 70 km2 has been estimated to survive and to have a measure of protection. Here it extends from the 2,000 m contour to the summit at 3,011 m alt. (Cheek, Onana & Pollard, 2000: 20). Elsewhere in the Cameroon Highlands, such as at Mt Kupe, Muanenguba, Bali-Ngemba, Ngel Nyaki and Dom, surviving patches of montane forest consists of only a few hectares, although an area of 40 km2 of forest has been recorded at Dutsin Dodo and Gangirwal mountain within the Gashaka Gumti National Park of Nigeria (H.M. Chapman, pers. obs., 2000).

The tree species diversity of the montane forest of the Cameroon Highlands is low (28 species, based on herbarium specimens, see Table 3) compared with submontane forest which has hundreds of species, and also in great contrast, montane forest contains few Cameroon Highland endemic tree species (only seven: 25% of the total, see Table 3). The majority of the canopy contains even fewer species. It was estimated that just ten species made up 90% of the montane forest canopy at Mt Oku and the Ijim Ridge, three of which are endemics (Cheek, Onana & Pollard, 2000: 20). The majority of montane tree species of the Cameroon Highlands are widespread in montane forest in Africa (Afromontane) occurring also east of the Congo Basin in the rift mountains of East Africa and several, such as Ilex mitis, extend north to Ethiopia and south to South Africa. The East African montane forest is more species-diverse, and only a subset of its species extend west to the Cameroon Highlands, and an even smaller subset, just seven species, extend even further west from the Cameroon Highlands, to the Guinea Highlands (Couch et al., 2019: 54).

The high altitudinal range of Deinbollia onanae is unrivalled west of the Congo basin by any other species of the genus. Elsewhere in Africa it is matched only by Deinbollia kilimandscharica Taub., of mountains from Ethiopia to Malawi, reported to achieve 2,250 m elevation in Tanzania (Davies & Verdcourt, 1998). Most species of the genus in tropical Africa are lowland forest shrubs, in the Cameroon Highlands only Deinbollia oreophila also occurs regularly at altitude over 800 m, and is largely confined to the submontane forest band being recorded from (880–)1,000–1,900(–2,050) m altitude where it is often relatively frequent (Cheek & Etuge, 2009a). We postulate based on their shared morphological characters that these two may be sister species (see results above) that have segregated between two adjacent altitudinally based vegetation types in a similar way to certain clades of bird species in the Cameroon Highlands such as the Turaco (Njabo & Sorenson, 2009). This hypothesis needs testing. It would most readily done by a comprehensive species-level molecular phylogenomic study of Deinbollia as has been achieved in several other genera, such as Nepenthes L.f. (Murphy et al., 2020).

The fruits of Deinbollia onanae are similar to those of other species of the genus, i.e., fleshy, indehiscent and large-seeded, suggesting that the now intermittent distribution of this species, along a line c. 570 km along peaks of the Cameroon Highland line, was likely due to dispersal in the gut by animals. Chimpanzees (Pan troglodytes ellioti) and putty-nose monkeys (Cercopithecus nictitans) are known to disperse the species at one location however these species do not cross from one forest patch to another, especially when as now these patches can be separated by tens of kilometres of secondary grassland. Formerly the range of Deinbollia onanae may have once been more continuous along the mountain range than today, but it was likely greatly reduced when forest was cleared for agriculture as reported above.

Conclusions

Such cases as Deinbollia onanae underline the urgency for publishing further discoveries while species still survive, since threats to such rare species are often clear and current, putting these species at high risk of extinction. About 2,000 new species of vascular plant have been discovered by science each year for the last decade or more (Cheek et al., 2020). Until species are delimited and known to science, it is more difficult to assess them for their conservation status and so the possibility of protecting them is reduced (Cheek et al., 2020). Documented extinctions of plant species are increasing, e.g., Oxygyne triandra Schltr. of Cameroon is now known to be globally extinct (Cheek et al., 2018b) as is Afrothismia pachyantha Schltr. (Cheek, Etuge & Williams, 2019). In some cases species appear to be extinct even before they are known to science, such as Vepris bali Cheek, once sympatric with Deinbollia onanae at Bali Ngemba (Cheek, Gosline & Onana, 2018), and elsewhere, Nepenthes maximoides Cheek (King & Cheek, 2020). Most of the >800 Cameroonian species in the Red Data Book for the plants of Cameroon are threatened with extinction due to habitat clearance or degradation, especially of forest for small-holder and plantation agriculture e.g., oil palm, following logging (Onana & Cheek, 2011). Efforts are now being made to delimit the highest priority areas in Cameroon for plant conservation as Tropical Important Plant Areas (TIPAs) using the revised IPA criteria set out in Darbyshire et al. (2017). This is intended to help avoid the global extinction of additional endemic species such as the Endangered Deinbollia onanae which will be included in the proposed IPA s of Mt Kupe, Bali Ngemba, Kilum-Ijim and Dom.

Most of the specimens cited in this paper were collected with the support of volunteers of Earthwatch Europe, Oxford and by our colleagues Kenneth Tah, Olivier Sene, Victor Nana, Verina Ingram, David Okebiro, Assefa, B. Gapta, H. Ndue, M. Kissimou, Rene Nfon, Stuart Cable, Ben Pollard and the late Martin Etuge. Drs Florence Ngo Ngwe, Eric Nana, Jean Betti Lagarde, the current and former directors, of IRAD-National Herbarium of Cameroon, Yaoundé, and their staff are thanked for expediting the collaboration between our two institutes. Nigerian specimens were obtained thanks to the Nigerian Montane Forest Project. Janis Shillito typed the manuscript. Xander van der Burgt made the map and the photo of the type specimen, and brought to light the overlooked Nigerian records. Emmanuel Barde Elisha of the Nigerian Montane Forests project helped provide information on local names and uses at Ngel Nyaki. Roy Gereau and David Kenfack are thanked for constructively reviewing an earlier version of this paper.

Additional Information and Declarations

Competing Interests

Author Contributions

Ethics

Field Study Permissions

Data Availability

New Species Registration

The authors declare there are no competing interests.

Martin Cheek conceived and designed the experiments, performed the experiments, analyzed the data, prepared figures and/or tables, authored or reviewed drafts of the paper, collected material, and approved the final draft.

Jean Michel Onana conceived and designed the experiments, performed the experiments, authored or reviewed drafts of the paper, collected material, and approved the final draft.

Hazel M Chapman analyzed the data, authored or reviewed drafts of the paper, collected material, and approved the final draft.

The following information was supplied relating to ethical approvals (i.e., approving body and any reference numbers):

Fieldwork was approved by the Institutional Review Board of the Royal Botanic Gardens, Kew, entitled the Overseas Fieldwork Committee (OFC 807-3).

The following information was supplied relating to field study approvals (i.e., approving body and any reference numbers):

Permission for fieldwork was given by IRAD (Institute for Agronomic Research and Development)-National Herbarium of Cameroon (000146/MINRESI/B00/C00/C10/C12).

The following information was supplied regarding data availability:

All raw data is already included Results section.

All specimens and associated metadata are cited within the paper (results section, following the standard conventions indicated in the methods section) as per best practice for botanical taxonomic publications, either as type specimens or under ”additional specimens”. Additionally, we also duplicate this information here for easy reader access:

Deinbollia onanae Cheek

Holotype: Cameroon, Mt Oku and the Ijim Ridge, Aboh to Tum, 2400 m alt., fl. 22 Nov. 1996, Etuge 3600 (holotype K000337729! Fig. 2, isotypes MO!, WAG0336084!, WAG0336083!, YA0057050!).

Additional specimens: CAMEROON. South West Region, Mt Kupe, near main summit, immature fr., 26 June 1996, Cable 3386 (K000197863!, YA!); North West Region. Bali Ngemba Forest Reserve, fr. April 2002, Onana 1600 (YA!); Mt Oku and the Ijim Ridge: above Laikom, st. 21 Nov..1996, Cheek 8709 (K000337728! YA!); Dom, Kinjinjang Rock, st. 25 Sept. 2006, Cheek 13436 (K000580433!; YA!); ibid. Forest Patch 1, fl. buds, 27 Sept. 2006, Cheek 13625 (K000580434!, MO!,US!, YA!); ibid., Javelong Forest, st. 29 April 2005, Pollard 1400 (K000580432!; YA!); Adamaoua Region, c. 120 km E of Ngaoundéré, 15 km NE of Belel, falls in Koudini River, alt. ± 1,200 m, fl. 4 Dec. 1964, W.J.J.O. & J.J.F.E. de Wilde, B.E.E. de Wilde-Duyfjes 4555 (K000593309!; K000593310!, WAG1269760! , YA). NIGERIA. Taraba State, Mambilla Plateau, Ngel Nyaki Forest Reserve, near camp, fr. 2 Dec. 2003, H.M. Chapman 481 (FHI, K!); ibid. female fl. 4 Dec. 2002, H.M. Chapman 484 (FHI, K!).

All specimens are herbarium specimens from the National Herbarium of Cameroon and Royal Botanic Gardens, Kew.

The following information was supplied regarding the registration of a newly described species:

Deinbollia onanae sp. nov.: 77215132-1

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
