# Peer review of "The montane trees of the Cameroon Highlands, West-Central Africa, with Deinbollia onanae sp. nov. (Sapindaceae), a new primate-dispersed, Endangered species"

_PeerJ, doi:10.7717/peerj.11036_

## Round 0.1 · original submission · Minor Revisions

Dear Dr. Cheek,

The reviewers think your manuscript is scientifically valid and well written. They also appreciated the information on the ecology of the species provided by the manuscript. They suggest some editing of the text that could further improve the manuscript. Please, respond point-to-point to the comments of reviewers to speed up the process of revision.

Once again, thank you for submitting your manuscript to PeerJ.

Sincerely,
Gabriele Casazza

·

Basic reporting

I think the paper would benefit from some careful editing of the text – including last paragraph- and I have made some comments below which I hope will help.

Basic reporting was good- the paper reads well with appropriate background and literature. I have some questions/ editorial comments:
L 107 What project? IPA’s needs a bit more explanation- UK’s PlantLife
L 109- don’t need surviving
L115 remove repeat ‘being’
L 125 add d to place (placed).
L 132 remove capital P from phylogenetics; add the to Sapindaceae?
L 142 five not 5
L 148 should this be highest known? Cameroon has had more recent intensive collecting than eg Nigeria.
L 190 remove the word was ..instead described in ..
L 189 Should YA (Yaunde) be defined, same with K (Kew) (same with other herbaria in L 192, L 200, L 201
L 212 remove brackets from (Cheek.. instead (2009)
L 225 – Table 1? (not table 1).
L 266 Deinbollia sp. And Deinbollia cf pinnata?
L 362 remove the ‘the’ following 1970s
L 367 -370 not very clear Rephrase- Moreover, observations of putty nose monkeys (Cercopithecus nictitans) feeding on Deinbollia at Ngel Nyaki (Gawaisa 2010) were of the primates in the crown of trees, so more likely…
L 386 could add- Putty nose monkeys also eat the leaves and flowers of Deinbollia (Gawaisa 2010; Hutchinson 2015)
L 379-386 could be written as follows: More numerous and so probably more effective at seed dispersal of Deinbollia are putty-nosed monkeys (Cercopithecus nictitans). Gawisa (2010) reported that at Ngel Nyaki Deinbollia fruit ranked third in preferred fruits of C. nictitans during February and March, and fifth in January. Hutchinson (2015) has shown that males in particular show a preference for Deinbollia seeds in the rainy season (Hutchinson 2014). Seeds are both swallowed, passing through the gut (average 2 per faecal sample) and sucked and spat by the putty-nosed monkeys (averaging 5 seeds per spitting event) (Chapman et al 2010). An experiment comparing germination time and success among Deinbollia seeds which had been defecated, spat and hand-cleaned found that gut passage had a significant beneficial effect on germination rates. A higher proportion of defecated seeds (60–70% ) germinated than spat (c. 40%) or hand –cleaned (c.35 %) seeds, and the defecated seeds germinated on average a few days earlier than non-defecated seeds (Chapman et al., 2010).

L 393 ….three months of the study (Matthesius et al )
L 411 412 Have you checked JDC 3550
First para of discussion, could you add the Memecylon described by Stone? Douglas Stone, in his research to improve the Memycylon genus phylogeny has said “ We now know that the Ngel Nyaki species should be placed in Memecylon (sect. Afzeliana Jacq.-Fél). HMC-744 clearly represents a new species, but further study of flowering & fruiting material is needed before it can be formally described and named. The new species is a small tree (about 15 metres tall) growing in Ngel Nyaki forest.”
L 499-504 There is a large block of forest in the Gotel Mountains, which are within the 6000km 2 Gashaka Gumpti National Park. InThere is approximately 40 square km’s of forest round Dutsin Dodo and Gangirwal mountain. In Chapman & Chapman 2001 this is not made clear as in this book the forest is described in sections based on species composition (and to some extent altitude). On a visit in 2000 we surveyed the forests which is where these data come from.
The data from Ngel Nyaki has all been funded under the umbrella of the Nigerian Montane Forest Project- so if that name could be included somewhere- maybe acknowledgements- would be appropriate.

References
Gawaisa G.S. (2010) The Role of Putty Nose Monkey (in forest regeneration of a montane forest ecosystem of Ngel Nyaki forest reserve, Taraba State, Nigeria) (Linnaeus, 1766). PhD thesis Federal Universitty of Technology, Yola, Nigeria /Nigerian Montane Forest Project.
Hutchinson K. (2015) Diet of Cercopithecus nictitans| and investigation into its potential to act as a surrogate disperser in disturbed Afromontane forests. MSc Thesis University of Canterbury, New Zealand

Experimental design

Hypothesis is clear and method of testing appropriate. Conclusions are well stated and concise.

Validity of the findings

I have full confidence in the validity of the findings. Data presented, sources explained and given.

Additional comments

In this contribution, the authors test the hypothesis that the tree species previously referred to as Deinbollia sp. 2 is a distinct, previously undescribed species, which they have named as D. onanae. The authors have assessed D. onanae as Endangered (IUCN 2012 criteria), and include valuable information on its ecology. I believe the authors have made a convincing case for this new species and provided ample, detailed evidence for there being a new species, likely a sister species of D. oreophila. This is a most valuable piece of work, worthy of publication in PeerJ. As explained in the paper, the flora and fauna of this diverse ecosystem are under intense threat from land clearance and fragmentation; it is essential that species be recognised and described before they go extinct- only by knowing what is there can a case be made for their conservation.
I think the paper would benefit from some careful editing of the text – including last paragraph- and I have made some comments below which I hope will help.

·

Basic reporting

The manuscript is very well written. The introduction is well-referenced and provides a thorough background on the genus Deinbollia, as well as the circumstances of the discovery of the new species. The three figures (line drawing, specimen scan and the distribution map) are of high quality, are all relevant and well labeled. However, the raw measurements of the plant parts are not provided.

Experimental design

The methods used by the authors are classic in taxonomy and are well described.

Validity of the findings

The conclusions are well stated. One of the merits of this manuscript is that it deviates from the classic taxonomic papers as it provides extensive details about the ecology of the new species.

Additional comments

The montane trees of the Cameroon Highlands, WestCentral Africa, with the Endangered, Deinbollia onanae sp. nov. (Sapindaceae), a new primatedispersed, Endangered species (#54299)

The authors describe a new species of Deinbollia already either recognized as a new species or otherwise misidentified in different parts of its distributional range. The authors provide a very detailed description of the new species and discuss its affinities with another congener that also occurs in high altitude. The conservation status of the new species is assessed following IUCN criteria and an excellent illustration of the new species is also included.
The manuscript is very well written. The introduction is well-referenced and provides a thorough background on the genus Deinbollia, as well as the circumstances of the discovery of the new species. The three figures (line drawing, specimen scan and the distribution map) are of high quality, are all relevant and well labeled. However, the raw measurements of the plant parts are not provided. The methods used by the authors are classic in taxonomy and are well described. The conclusions are well stated. One of the merits of this manuscript is that it deviates from the classic taxonomic papers as it provides extensive details about the ecology of the new species.

The following suggestions and comments could further improve the manuscript:
Line 69 & 70: “It is postulated that this new species is in a sister relationship with Deinbollia oreophila”. The authors should simply say that the new species is morphologically close to Deinbollia oreophila.
The “sister relationship” cannot be postulated on the sole basis of morphology. Carapa oreophila and C. angustifolia that are also the only two species of the genus occurring in altitude in the Cameroon line occur in different clades of the phylogeny.

Line 125: The genus Deinbollia Schum. & Thonn. is traditionally placed in the tribe Sapindeae DC

Line 192: …were seen by one or more both authors

Line 239 &240: “The affinities of Deinbollia sp. 2 may be with the recently described D. oreophila since this species also occurs at altitude in the Cameroon Highlands and both species share numerous”.
See above. Sympatric species are not necessary sister species

Line 341: … Jean-Michel Onana, currently Senior Lecturer in Botany at University of Yaoundé I

Line 146: … Taxonomic Checklist of the Vascular Plants of Cameroon Cameroon (Onana, 2011).

Line 359 & 362: We consider that many and probably most of the smaller of these numerous stems may not be the usually infrequent D. onanae, but the much smaller (0.8–361 3(–5) m tall) D. oreophila which at this altitude, over the border in Cameroon, is vastly more frequent in submontane forest (Cheek & Etuge 2009)
The diameter size class distribution of D. ananae has a reverse J shape at Ngel Nyaki which simply means that this species regenerates well in this locality. Speculating that the smaller stems of this species may belong to a separate species is misleading. We all know that taxonomists are mostly interested in fertile herbarium specimens that are often collected from mature trees, ignoring all juveniles of the same species in the understory.

Line 162 – 165: In contrast, the 1970s the 1 ha enumeration plot at Ngel Nyaki (Chapman & Chapman 2001: 25–26) yielded five stems of “Deinbollia sp.” in the C strata (understorey trees 7–13 m high) with diameter at 1.3 m above the ground exceeding 14.5 cm, of which two exceeded 28 cm and one 40 cm
There is no contrast here. The Ngel Nyaki plot includes all stems with dbh > 1 cm. In this plot, there were 3078 trees, of which only 95 had a diameter > 14.5 cm (4.6 individuals per ha) versus 5 individuals/ ha enumerated by Chapman & Chapman.

Line 367 – 368: observations of animals feeding on Deinbollia at Ngel Nyaki have been made using binoculars of primates trained on the crowns of trees so are,
Not sure what this means

Line 506: “The tree species diversity of the montane forest of the Cameroon Highlands is low (28 species,”
Is this estimate of 28 species based on inventory plots (which dbh cutoff?) or herbarium collections?

·

Basic reporting

no comment

Experimental design

no comment

Validity of the findings

no comment

Additional comments

The article is scientifically sound, follows a perfectly standard format for this kind of taxonomic presentation, and leaves no doubt that the species described is indeed new to science and worthy of recognition. The suggested taxonomic affinities are well supported and testable by further research. There are no systematic flaws or areas for general improvement. I have made mostly very small suggestions on details of the text in the form of sticky notes in the attached PDF. Most are completely self-explanatory, but I will make specific comments on a few:

1. The words "the Endangered," should be removed from the title because they are redundant, with "Endangered species appearing at the end of the title.

2. In the formal diagnosis following line 255, the highlighted series of character states are identical to those with which they are being contrasted; these must be replaced by the correct character states. In general, the diagnosis could be rearranged to place the contrasting states closer to those with which they are being contrasted.

3. The Area of Occupancy is given in the conservation assessment on line 444. The Extent of Occurrence should also be given, even if it is larger than the threshold value for any threatened category. Severe habitat fragmentation was apparently used to assess this species as Endangered despite the existence of six threat-defined locations; the connection of habitat fragmentation with the resulting assessment should be made clearer.

With these minor points addressed the manuscript will be ready for publication.

Roy E. Gereau
Missouri Botanical Garden

---

## Round 0.2 · Minor Revisions

Dear Dr. Cheek,

The manuscript was strongly improved according to the suggestions of the three reviewers. Because some mistypes still occur through the text, I ask you to check carefully the text before final acceptance. Moreover, a large part of “Distribution & ecology” chapter seems not to be your result but more a discussion based on results of other studies. Please, consider moving part of this chapter to the Discussion.

Some mistypes:
L143-144: There are several "and" in the list
L149, 219, 345, 796 and 799: May be “Flore du Cameroun” instead of “Flore Du Cameroun”
L189: The authors are now three. May it be part of the old version?
L336: You usually use Ngel Nyaki and not Ngel-Nyaki. Please uniform the name spelling
L361: “We consider that many and probably most of the smaller of these numerous stems may” The sentence is incomplete. Complete or delete it.
L377: There is an issue of incompatibility between operating systems or software. There is a square between Nigerian and Cameroon. Please, put the correct symbol here
L451: Please uniform the spelling of Geocat (previously GeoCAT).

Once again, thank you for submitting your manuscript to PeerJ and compliments on your interesting article.

Sincerely,
Gabriele Casazza

---

## Round 0.3 · accepted · Accept

Dear Dr. Cheek,

I am very pleased to inform you that your paper "The montane trees of the Cameroon Highlands, West-Central Africa, with Deinbollia onanae sp. nov. (Sapindaceae), a new primate-dispersed, Endangered species " is accepted for publication in the PeerJ. Congratulations!

Thank you for submitting your work to PeerJ.

Sincerely,

Gabriele Casazza